# Beetle luciferases with naturally red- and blue-shifted emission

César Carrasco-López[1] , Juliana C Ferreira[1], Nathan M Lui[1] , Stefan Schramm[1], Romain Berraud-Pache[2], Isabelle Navizet[2] , Santosh Panjikar[3,4] , Pance Naumov[1], Wael M Rabeh[1]

The different colors of light emitted by bioluminescent beetles that use an identical substrate and chemiexcitation reaction sequence to generate light remain a challenging and controversial mechanistic conundrum. The crystal structures of two beetle luciferases with red- and blue-shifted light relative to the green yellow light of the common firefly species provide direct insight into the molecular origin of the bioluminescence color. The structure of a blue-shifted green-emitting luciferase from the firefly *Amydetes vivianii* is monomeric with a structural fold similar to the previously reported firefly luciferases. The only known naturally red-emitting luciferase from the glow-worm *Phrixothrix hirtus* exists as tetramers and octamers. Structural and computational analyses reveal varying aperture between the two domains enclosing the active site. Mutagenesis analysis identified two conserved loops that contribute to the color of the emitted light. These results are expected to advance comparative computational studies into the conformational landscape of the luciferase reaction sequence.

## Introduction

The dazzling flashes of bioluminescent light communicated by fireflies have inspired scientists, writers, artists, and laymen for centuries. Beyond its visual appeal, bioluminescence is an irreplaceable bioanalytical tool for in vivo imaging, monitoring of cell proliferation, studies into protein folding and secretion, environmental research, and food quality control. At the core of this natural phenomenon lies a fundamental process of energy transduction by luciferase enzymes that convert the chemical energy stored within the ground-state substrate (luciferin) to an excited, emissive state of the product (oxyluciferin) by a spin-forbidden process (Johnson & Shimomura, 1972; DeLuca, 1976). Although beetle luciferase systems share identical substrates and chemical reaction sequence, they emit a range of different colors from yellow green

($\lambda_{max} \approx 560$ nm), which is typical for common firefly species, such as the North American firefly *Photinus pyralis* ($G_{Pp}$) and the Japanese firefly *Luciola cruciata* ($G_{Lc}$), to orange and even red ($\lambda_{max} = 590–623$ nm) from certain click beetles and railroad worms (Viviani et al, 1999, 2011; Ugarova & Brovko, 2002). The molecular origin of the different colors of light emitted by different luciferases remains the most elusive mechanistic aspect of this photochemistry and has been continuously debated since the 1970s. In the absence of structural information on WT luciferases that emit light other than green, several mechanisms have been advanced and subsequently refuted. Current mechanistic models that are based on experimental and computational analysis on available green-emitting luciferase structures remain inconclusive (Hosseinkhani, 2011).

Here, we describe the first crystal structures of two rare WT luciferases from Brazilian beetles that emit light with exceptional colors; a luciferase from the head lanterns of the glow-worm *P. hirtus* (Coleoptera: Phengodidae), the only known luciferase that naturally emits red light ($\lambda_{max} = 623$ nm; $RE_{Ph}$), and a green-emitting luciferase from the firefly *A. vivianii* (Coleoptera: Lampyridae) that displays a blue-shifted emission relative to common firefly luciferases ($\lambda_{max} = 538$ nm at pH 8; $GB_{Av}$) (Viviani et al, 1999, 2008, 2011). Biochemical and structural analyses of the two luciferases, combined with computational modeling, provide the best insight yet into the relationship between the structure and color of light emitted by beetle luciferases.

## Results

### Structural determination of the red-emitting luciferase from *P. hirtus*

The crystal structure of WT $RE_{Ph}$ was determined at low resolution by molecular replacement from two different crystal forms in the space groups $P1$ and $P3_121$ at resolution of 3.05 Å and 3.60 Å, respectively (Table S1). Both crystal forms presented good-quality electron density maps, which were improved by the non-crystallographic symmetry of both unit cells (Fig S1A and B). Unlike previously reported luciferases that are exclusively monomeric (Conti et al, 1996; Franks et al, 1998;

[1]New York University Abu Dhabi, Abu Dhabi, United Arab Emirates   [2]Laboratoire Modélisation et Simulation Multi Echelle, MSME UMR 8208 CNRS, Université Paris-Est, Marne-la-Vallée, France   [3]Australian Synchrotron, Clayton, Australia   [4]Department of Biochemistry and Molecular Biology, Monash University, Melbourne, Australia

Correspondence: wael.rabeh@nyu.edu

Nakatsu et al, 2006; Auld et al, 2010; Cruz et al, 2011; Sundlov et al, 2012; Kheirabadi et al, 2013; Branchini et al, 2017), in the $P3_121$ crystal form, $RE_{Ph}$ exists as tetramer, although in the $P1$ crystal form it is an octamer in the asymmetric unit (Figs 1A and B, S2A, and B). The N-terminal domains in the octamer core structure are assembled as a tetramer of dimers and packed over dimer and tetramer interfaces, and the C-terminal domains point outward. This assembly accounts for the structural flexibility and increased thermal motion of the C-terminal domains which is apparent from the residual electron density. The inability to model the C-terminal domain of firefly luciferases in certain crystal conditions, as result of its high flexibility, has been previously shown (Auld et al, 2010; Thorne et al, 2012; Kheirabadi et al, 2013). Only one out of the four C-terminal domains was observed in the density maps of the $P3_121$ crystal form and none of the eight C-terminal domains of the $RE_{Ph}$ octamer could be resolved in the $P1$ crystal form (Supplementary Note 1). The $RE_{Ph}$ structure is consistent with the α/β-fold of other beetle luciferases and the substrate-binding pocket in each of the monomers is located between the larger (N-terminal) and smaller (C-terminal) domains. A displacement of the C-terminal domain can open and close the active site in a conformational rearrangement that is purportedly triggered by binding of the substrates (Nakatsu et al, 2006).

In the structure of $RE_{Ph}$, each dimer within the octamer is stabilized by multiple hydrogen bonds between amino acid residue R11 from one of the monomers and Y26, Y30, and N179 from the other monomers, which extend over a $C_2$ axis across the dimer interface (Fig 1C). The interactions across the dimer interface are strong

electrostatic interactions with two contact points contributed by R11 from each monomer (Fig 1C). With the low resolution of the $RE_{Ph}$ structure, site-directed mutagenesis clearly confirms these interface interactions, where the single mutation (R11A) was sufficient to disrupt the octamer of $RE_{Ph}$ and resulted exclusively monomers in solution (Fig S2C and D). On the other hand, the hydrophobic interactions between M152, Y153, and F162 from two dimers contribute to the weak overall interactions between the dimers over the tetramer interface (Fig 1D). As confirmed by mutagenesis and size-exclusion chromatography, the interactions across the tetramer interface are weaker relative to the interaction of the individual monomers over the dimer interface. The WT $RE_{Ph}$ exists as a tetramer in solution (Supplementary Note 2). Consequently, mutations on the tetramer interface, which include single (F162A) and double (Y153A and F162A) mutants, produced only dimers in solution (Fig S2C). Overall, the emission of WT $RE_{Ph}$ remains unaffected by interface mutations that result in fragmentation of the $RE_{Ph}$ octamer into dimers and monomers; we therefore conclude that the red emission of $RE_{Ph}$ is not a result of its quaternary structure. Instead, the red light is intrinsic to the structural fold of the $RE_{Ph}$ monomer and originates from the specific packing and microenvironment of its active site.

## Structural determination of the blue-shifted green emission luciferase from *A. vivianii*

To identify structural features that are important in the color tuning mechanism, the crystal structure of $GB_{Av}$ luciferase with a blue-shifted

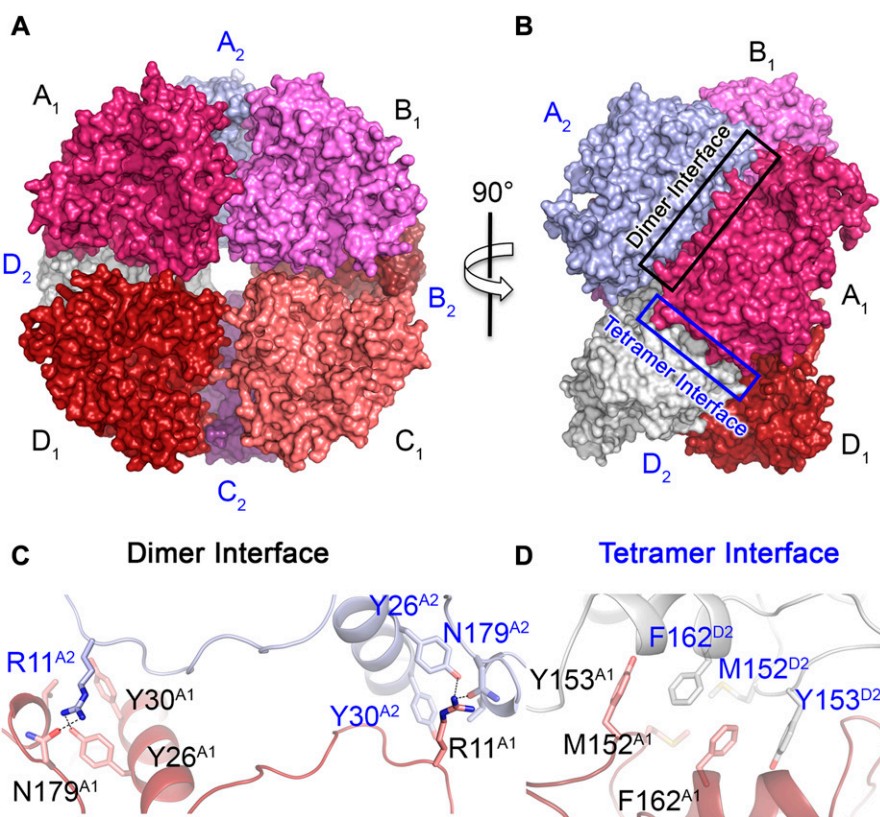

**Figure 1. Crystal structure of the naturally red-emitting luciferase from *P. hirtus* ($RE_{Ph}$).**
**(A)** Front view of the partial octameric conformation found in the $P1$ crystal form (only the N-terminal domains were observed in the electron density maps). **(B)** Lateral view of the octamer that highlights the surface interactions between the monomers. **(C)** Close-up view and interactions across the dimer interface between monomers A1 (red) and A2 (blue). The interacting residues are shown as stick models with matching colors as monomers A1 and A2 in panels (A, B). The two surfaces are related to each other by a $C_2$ axis. The broken lines show the interactions between residues R11, N179, and Y26. Mutation R11A disrupted the octamer to give monomers in solution. **(D)** Close-up view of the tetramer interface across the dimers, which are assembled as an octamer (monomer A1 is dark red and monomer D2 is white). The interactions between the two surfaces are predominantly hydrophobic interactions between Y153, M152, and F162 from both dimers. Similar to the dimer interface, the surfaces at the tetramer interface are related by a $C_2$ axis. Single and double mutations at residues Y153 and F162 generated exclusively dimers in solution.

green emission of $l_{max}$ = 538 nm at pH 8 was determined for comparison with the RE$_{Ph}$ (Fig 2B and C). The emission of GB$_{Av}$ is at higher energy relative to the well-studied luciferases G$_{Lc}$ (Nakatsu et al, 2006) and G$_{Pp}$ (Conti et al, 1996) that emit green-yellow light. The GB$_{Av}$ crystals diffract to a resolution of 1.9 Å when free of substrate and crystallize in a space group $P2_12_12_1$ with two independent GB$_{Av}$ monomers in the asymmetric unit (Fig S1E and F and Table S1). Although the N-terminal domains of the two GB$_{Av}$ molecules are very similar (root-mean-square deviation [RMSD] value of 0.1 Å calculated on all backbone atoms), their C-terminal domains have different conformations with an RMSD value of 2.1 Å (Fig S3A and B). Thus, although the luciferase in both molecules can be considered as being in its "open" conformation, the luciferase with the smaller aperture is maybe an intermediate between the "open" and "closed" conformations (Figs 2B, S3A, and B), the latter being attributed to the structure of G$_{Lc}$ in complex with the reaction products (Nakatsu et al, 2006). Indeed, a superposition of the N-terminal domains of the "open" conformations of GB$_{Av}$ and RE$_{Ph}$ with the green-emitting luciferase G$_{Lc}$ in complex with oxyluciferin and AMP (Protein Data Bank

[PDB] code: 2D1R), which is in its "closed" state (Nakatsu et al, 2006), indicates a movement of the C-terminal domain of about 10 Å to 30 Å between the two states to open/close the active site (Fig S4). The pronounced flexibility of the C-terminal domain of GB$_{Av}$ was further examined by classical molecular dynamics simulations in which the C-terminal domain required 5 to 15 ns to shuffle between the two conformations. Together with the structural data, these results confirm the pronounced mobility of the C-terminal domain of beetle luciferases, which is capable of reversible opening and closing of the active site through two catalytic conformations during the bioluminescence reaction. The two catalytic conformations are stimulated by rotation on the C-terminal domain of firefly luciferases (Sundlov et al, 2012). Notably, the structures of both GB$_{Av}$ and RE$_{Ph}$ have wider openings than G$_{Pp}$ luciferase devoid of substrates or products (PDB codes: 1LCI and 5DV9; Figs 2A and B, and S4A) (Conti et al, 1996; Wu et al, 2017). The aperture of the active site with an angle of ~125° between axes crossing the center mass of each domain ($P3_121$ crystal form) shows that RE$_{Ph}$ is the most open conformation

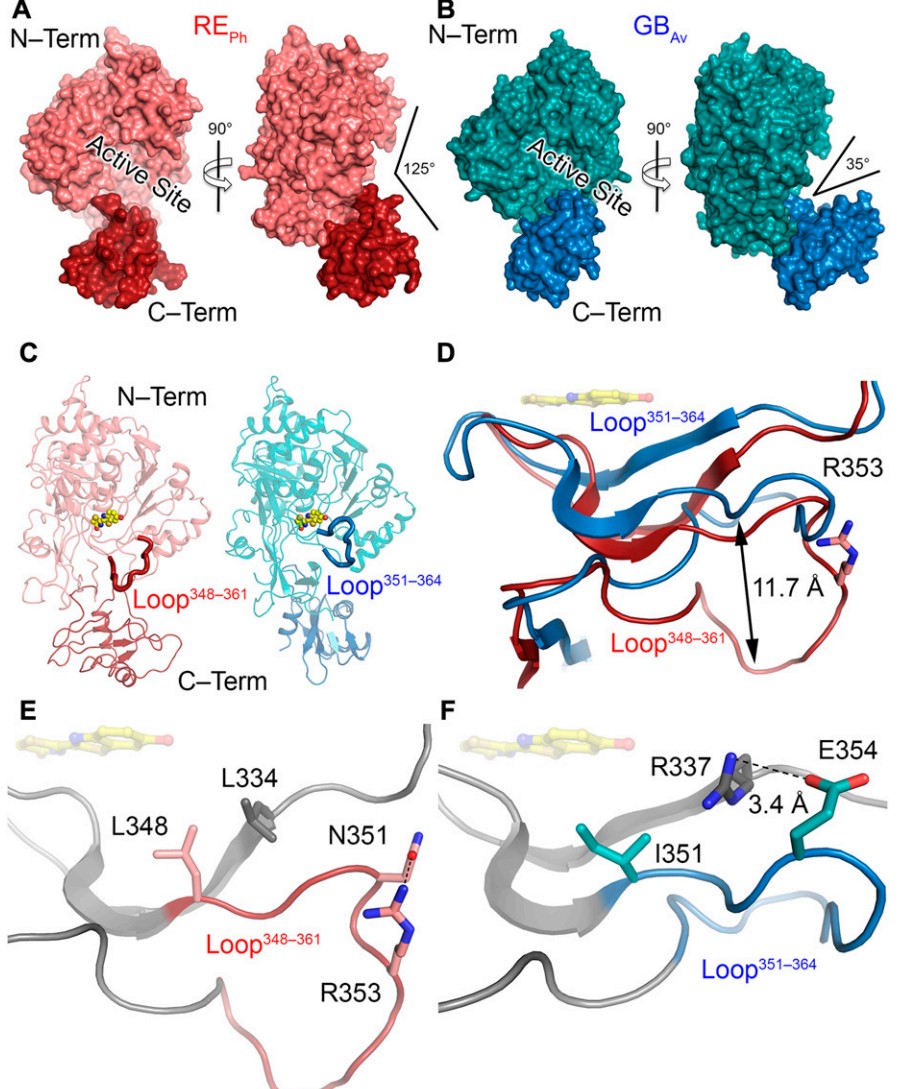

Figure 2. **Analysis of the structures of red-emitting RE$_{Ph}$ and blue-shifted green-emitting GB$_{Av}$ luciferases.** **(A)** Full-length structure of molecule B, the only monomer with a complete C-terminal domain of RE$_{Ph}$ in the $P3_121$ crystal form (the structures of the other three monomers in this crystal lack the C-terminal domain, which could not be observed in the difference electron density maps). The conformation of RE$_{Ph}$ has the largest aperture between the N-terminal ("N-Term") and C-terminal ("C-Term") domains among the luciferases with known crystal structures. **(B)** Structure of one of the two monomers in the asymmetric unit of GB$_{Av}$ (the monomer of GB$_{Av}$ with a larger aperture is shown here; the angle of the aperture of the other monomer is 30°). The structural packing of both GB$_{Av}$ monomers is less open relative to RE$_{Ph}$. The RMSD value of the superimposed monomers is 0.22 Å based on all α-carbons in the structure (the deviations were prominent in the C-terminal domain, with RMSD of 2.1 Å). Identical conformations were found for the N-terminal domain, with RMSD based on the α-carbons of 0.08 Å. **(C)** The loose packing of loop$^{348-361}$ (red) in the N-terminal domain of RE$_{Ph}$ relative to the tight packing of loop$^{351-364}$ (blue) in the N-terminal domain of GB$_{Av}$ (see Fig S6). **(D)** The shift of loop$^{348-361}$ (red) in RE$_{Ph}$ relative to loop$^{351-364}$ (blue) in GB$_{Av}$ by superposition of the two monomers based on the α-carbons of the whole structures (Fig S6A). R353 (pink sticks) is the only known insertion in the RE$_{Ph}$ sequence. **(E)** Natural substituted residues found in loop$^{348-361}$ (pink) of RE$_{Ph}$ close to the active sites are shown in gray and pink sticks. **(F)** The relatively conserved residue counterparts in loop$^{351-364}$ (blue) of GB$_{Av}$ are shown in gray and blue sticks. In panels D, E, and F, the reaction product (oxyluciferin, shown with yellow sticks) is shown by superimposing the structures reported here with the structure of G$_{Lc}$ in complex with oxyluciferin (PDB code: 2D1R).

among the known luciferase structures, which corresponds to the high mobility of its C-terminal domain. This feature indicates that the flexibility of the C-terminal domain may play a role in tuning the color of light emitted.

## Loop[351–364] is important for the green and red emission of GB$_{Av}$ and RE$_{Ph}$

Detailed analyses of the structures of RE$_{Ph}$, GB$_{Av}$, and G$_{Lc}$ revealed multiple amino-acid residues that may be directly involved in determining the color of emitted light (Supplementary Note 3). The most remarkable structural feature is the conserved loop[351–364] in GB$_{Av}$ (which corresponds to loop[348–361] in RE$_{Ph}$) that is located in the N-terminal domain at the edge of the active site (Figs 2C and D, and S5). In addition, the presence of the only known insertion in beetle luciferases (R353 in RE$_{Ph}$; Fig S5), the proximity of loop[351–364] to the active site and several key substitutions around the benzothiazole-binding region, which were previously found to have an impact in the color tuning (Viviani et al, 2007), drove us to investigate the relevance of loop[351–364] in the color-tuning mechanism. The loop[348–361] in RE$_{Ph}$, albeit at low resolution, was modeled in the electron-density map (Fig S1C). In GB$_{Av}$, loop[351–364] is tightly held by strong ionic interactions between E354 on the loop and R337 on the N-terminal domain (Figs 2F, S6, and S7). Residue K358 on the opposite side of the loop also interacts with D427 and D429 from the terminal section of the N-terminal domain (for figure clarity, these interactions are not shown in Fig 2F). The ionic interactions on both sides of loop[351–364] fix its position relative to the enzyme backbone. This stability is reflected in the low RMSD value for the loop backbone atoms of only 0.3 Å between GB$_{Av}$ and G$_{Lc}$. These interactions are absent in RE$_{Ph}$ owing to the replacement of R337 and E354 in GB$_{Av}$ with L334 and N351 in RE$_{Ph}$, respectively (Figs 2E, S5, and S6A). The absence of these strong interactions with the enzyme core (represented by R337) increases the mobility of analogous loop[348–361] in RE$_{Ph}$, as reflected in the higher RMSD value of 2.3 Å for the respective atoms between the RE$_{Ph}$ monomers. Notably, RE$_{Ph}$ is also the only beetle luciferase with an additional residue, R353, in loop[348–361] (Figs 2E and S5). Although insertion of R353 in green-emitting luciferases including, GB$_{Av}$ (Fig 3D) and Table S2), red-shifts the color that is emitted (Tafreshi et al, 2007; Alipour et al, 2009), its deletion from the red-emitting RE$_{Ph}$ as demonstrated here and previously does not affect the red emission (Viviani et al, 2007).

Additional biochemical and computational analyses were performed to assess the effects of R337 on the emission color because our and others' (Viviani et al, 2007; Viviani et al, 2016) mutation experiments indicated that it has an important role in determining the emitted color (Fig 3C and D and Supplementary Notes 3 and 4 and Tables S3, S4, and S5). Mutation of R337L in green-emitting GB$_{Av}$ red-shifted its emission by 42 nm, from 538 to 580 nm (Fig 3B). The theoretical calculations reproduced the observed trend and estimated a red shift of ≈60 nm, from 535 to 577 nm (Table S4, snapshot 4). A second mutation, I351L, did not affect the emission energy of R337L, which shows the dominating effect of R337 (Table S3). These results confirm that the interactions of R337 with loop[351–364] in GB$_{Av}$ are critical for the emission of green light. We anticipate that this effect is more general for green-emitting luciferases and an analogous mutation that disrupts these interactions would shift their green emission ($\lambda_{max} \approx 560$ nm) and thereby decrease the

emitted energy ≈ 40–60 nm to generate emission of red light (≈600–620 nm). Thus, it appears that the absence of strong interactions of L334 in red-emitting luciferase RE$_{Ph}$ (analogous to R337 in GB$_{Av}$) is critically important for its red emission. In support of this hypothesis, replacement of L334 (mutation L334R) in red-emitting RE$_{Ph}$ blue-shifted the light of this luciferase by 18 nm, from 623 to 605 nm without a significant change in light intensity (Fig 3A). Similarly, the double mutant L334R/L348I of RE$_{Ph}$ blue-shifted emissions from 623 to 600 nm, although the single mutation L348I did not affect the emission maximum, which confirms that L334 in RE$_{Ph}$ is the key residue that determines the low energy of its emission. Nevertheless, these mutants still emit red light. We hypothesize that synergistic effects that involve residues outside the active site could stabilize the closed state and shift the emission further to the green.

## Loop[523–530] only contributes to the green emission of GB$_{Av}$

A second highly conserved loop in beetle luciferases at the end of the C-terminal domain, loop[523–530] in GB$_{Av}$ (loop[521–528] in RE$_{Ph}$), is also relevant to the aperture of the enzyme (Figs S4 and S5). Despite the low resolution of the RE$_{Ph}$ structure, the high-quality electron density map allowed to model this loop region in the $P3_{1}21$ crystal form (Fig S1D), where rotation and displacement of loop[523–530] by ~10 Å closes the active site and facilitates interactions of key residues with the substrates. In the closed state of G$_{LC}$, K526 interacts with T292 and D468 of the N- and C-terminal domains, respectively (Fig S4E). Upon closure of the active site, T527 and K529 of GB$_{Av}$ are shifted to interact with the substrate in accordance with their catalytic roles in the bioluminescence reaction (Branchini et al, 2000, 2004, 2005). To assess the role of the conformation of the C-terminal domain on the emission from the excited state of the product, a complementary umbrella classical molecular dynamics simulation was performed on GB$_{Av}$ in which K524 of the C-terminal domain was constrained to interact with the backbone of G311 of the N-terminal domain to close the active site in GB$_{Av}$ (Figs S5, S8, and S9 and Supplementary Note 4). The results revealed that rotation of the C-terminal domain between the open and closed states is not related to energetic barrier. Moreover, the bioluminescence emission calculated for the closed state (530 ± 10 nm) is in strong agreement with the experimental value of 538 nm (Table S6). Thus, the presence of loop[523–530] inside the active site of the luciferase is required for the natural bioluminescence color of WT GB$_{Av}$.

We were able to discern with sufficient accuracy the structure of loop[523–530] in both independent molecules in the crystal of GB$_{Av}$, in which the open conformation is stabilized by interactions between T527 and K529 of loop[523–530], and D422 and D436 of the N-terminal domain (Fig S4C). The structural integrity of loop[523–530] is maintained by interaction between K524 and the backbone of G528. The mutant K524A in loop[523–530] of GB$_{Av}$ exhibited a red shift of 12 nm with minimum decrease in its thermodynamic stability (Figs 3C and D, S10B, E, F, and G–I, and Supplementary Note 3). The emission of the same mutant (K522A) of RE$_{Ph}$ remained unaltered, even though its thermodynamic stability decreased (Figs 3C, S10A, C, D, and G–I). With the exception of T525A in RE$_{Ph}$, which retained ≈ 40% of its emission intensity, other mutants in loop[523–530], which include T527A and K529A of GB$_{Av}$, and T525A and K527A of RE$_{Ph}$, suppressed the WT enzyme emission completely.

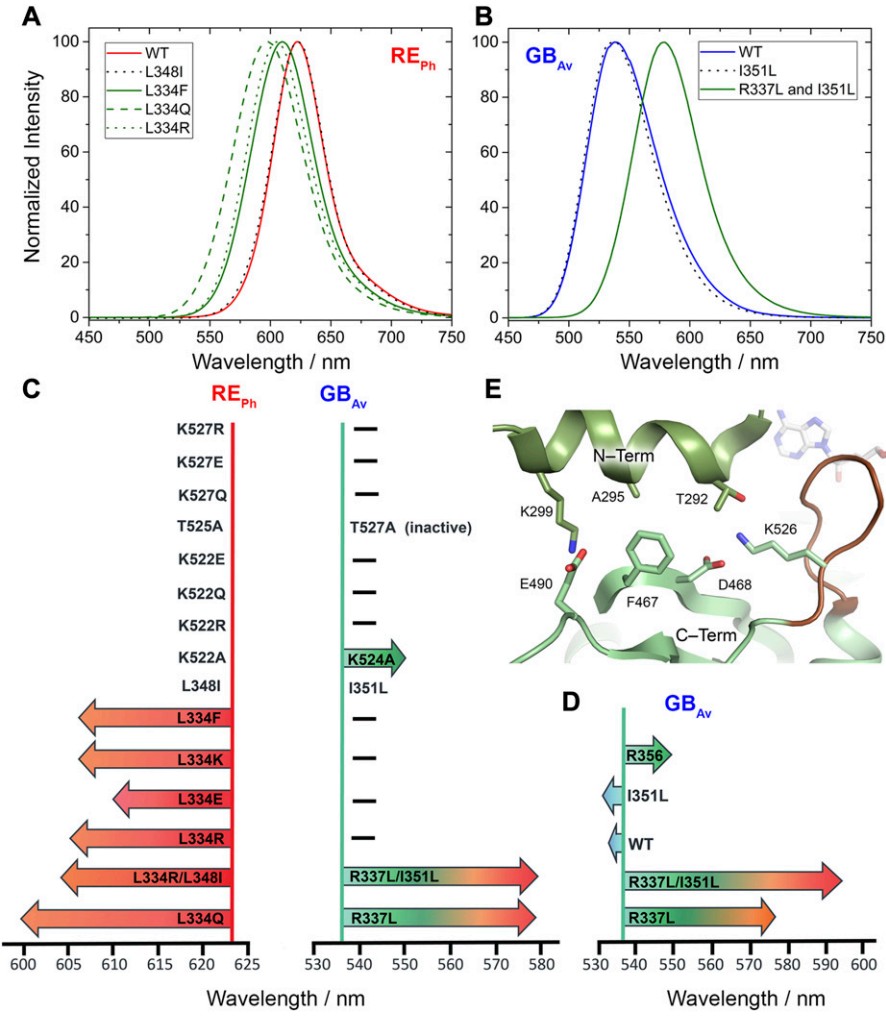

**Figure 3. Normalized bioluminescence emission spectra and kinetics of WT and mutants of luciferases RE$_{Ph}$ and GB$_{Av}$.**
**(A)** At pH 8.0, the emission from WT RE$_{Ph}$ with $\lambda_{max}$ = 623 nm was blue-shifted between 600 and 610 nm in mutants of L334. **(B)** At pH 8.0, GB$_{Av}$ emission at $\lambda_{max}$ = 538 nm for the WT enzyme was red-shifted to 580 nm in double mutant R337L/I351L. Single mutant I351L did not alter the emission of GB$_{Av}$. **(C, D)** Schematic of the experimental (C) and calculated (D) data for mutation-induced shifts of emissions of GB$_{Av}$ and RE$_{Ph}$ (Tables S2, S3, S4, and S5). The vertical y-axis is set at the WT emission of GB$_{Av}$ or RE$_{Ph}$. Each arrow represents a mutant (labeled inside the arrow) that shifts the color from the WT emission and points in the direction of change of the emitted color. The tip of the arrow is a qualitative representation of the color shift and the maximum emission wavelength. The mutations that did not affect the color emitted by the WT luciferase are labeled immediately next to the y-axis. **(E)** Residues at the interface (green sticks) between the N- and C-terminal domains in the closed conformation of G$_{Lc}$. Previous mutations on residues at this interface (E490 and F467) and mutant K524A in GB$_{Av}$ (K526 in G$_{Lc}$) reported here red-shifted the color between 10 and 15 nm.

The relevance of the C-terminal domain in the color-tuning mechanism was further explored by introducing mutation K524A, which is distant from the active site in both the open and closed conformations (Fig 3E). The slightly decreased thermodynamic stability of this mutant indicates minor destabilization of the interactions of K524 with T290 and D466 (as seen in the structure of GB$_{Av}$), which are at the interface between the N- and C-terminal domains in the closed state (Fig S4E and Supplementary Note 3). This result is similar to the red shift in emission of other mutations that have been introduced previously in the C-terminal domain and are located on the same interface (Fig 3). This interface between the N- and C-terminal domains includes polar and hydrophobic interactions that, when disrupted, are known to shift the color to red by ≈10–15 nm without affecting the general fold of the enzyme (Modestova & Ugarova, 2016).

## Discussion

The structural determination of the only naturally red-emitting RE$_{Ph}$ has, for the first time, revealed an oligomeric structure for a beetle luciferase, whereas others are all exclusively monomeric. Although oligomerization is possible in solution, it is not critical for the emission of red light; instead, the low emission energy is inherent to the structure of the monomer and the microenvironment of the active site. Although co-crystals with either substrate could not be achieved, the structures of RE$_{Ph}$ with red emission and GB$_{Av}$ with blue-shifted green emission provide insight into the effects that conformational changes and active site microenvironment have on the color of the light emitted by luciferases. Furthermore, two conserved segments, loop[351–364] in GB$_{Av}$ and loop[348–361] in RE$_{Ph}$, were identified by mutagenesis to have a profound effect on the emission of both enzymes. The strong interactions of R337 with loop[351–364] in GB$_{Av}$ are important for its green emission. Similarly, mutations that altered the interactions with loop[348–361] blue-shift the emission of RE$_{Ph}$ from 623 nm, for the WT, to 605 nm in the L334R mutant. The second conserved segment, loop[523–530] in GB$_{Av}$ or loop[521–528] in RE$_{Ph}$, was found to be important for the emission of GB$_{Av}$ only, and substitutions in loop[521–528] did not alter the red emission of RE$_{Ph}$. These results provide direct insight into the molecular origin of the diverse colors emitted by different beetle luciferases and are the key to solving one of the most difficult conundrums in bioluminescence research.

# Materials and Methods

## Materials

Unless mentioned otherwise, the chemicals were from Sigma-Aldrich. The synthesis of full-length genes and oligonucleotide DNA primers, DNA sequencing, and, in few cases, site-directed mutagenesis were carried out by GenScript USA Inc. Restriction enzymes were purchased from New England Biolabs Inc., and KOD DNA polymerase and dNTP mix were from EMD Millipore.

## Cloning, expression, and purification

The genes of the natural red-emitting luciferase from *P. hirtus* (RE$_{Ph}$; railroad worm) and the green-emitting luciferase with blue-shifted emission (GB$_{Av}$) from *A. vivianii* were synthesized by GenScript (GenScript USA Inc.). The DNA sequences, based on the previously reported proteins sequences (Viviani et al, 1999, 2011), were optimized for *Escherichia coli* expression and designed to be subcloned into a pET26b-derived bacterial expression vector that contains an N-terminal domain SUMO fusion protein tag (Champion pET SUMO system, Thermo Fisher Scientific) by using NheI and XhoI as restriction sites. Further subcloning was performed into selected pET28-derived systems by using NheI–HindIII restriction sites (RE$_{Ph}$) and BamHI–HindIII (GB$_{Av}$). The best overexpression results were obtained for GB$_{Av}$ cloned into the pET28a+ system (Novagen), whereas the RE$_{Ph}$ expression was improved by using the pET28 MHL system (provided by the Structural Genomics Consortium). Most of the point mutations were introduced by site-directed mutagenesis (Edelheit et al, 2009), whereas others were ordered from GenScript (all point mutations used to destabilize the octamer).

The luciferase constructs were introduced by transformation into *E. coli* BL21-CodonPlus-RIL (Stratagene). A single colony was used to inoculate Luria broth that contains kanamycin (100 mg/l) and chloramphenicol (50 mg/l) and grown overnight at 37°C with vigorous shaking (preinocules). The inoculated cultures (usually 4–6 liters for each construct) were grown at 37°C until the $A_{600}$ reached 0.2. At this point, the temperature was lowered to 28°C until an $A_{600}$ of 0.3 was achieved, and the luciferases expression was then induced overnight (at 15°C) by adding IPTG (0.2 mM). The cells were harvested by centrifugation at 12,000 *g* for 10 min in an Avanti J26-XPI centrifuge (Beckman Coulter Inc.), then resuspended in lysis buffer (Tris [100 mM], pH 7.8, NaCl [500 mM], glycerol [10%], imidazole [5 mM], βME [3 mM], and protease inhibitor cocktail from Sigma-Aldrich: P8849), lysed by sonication on ice, and centrifuged again at 40,000 *g* rpm for 30 min at 4°C. The supernatants were loaded at a rate of 1 ml/min onto a ProBond Nickel-Chelating Resin (Life Technologies) previously equilibrated with binding buffer (Tris [100 mM], pH 7.8, NaCl [500 mM], glycerol [10%], imidazole [5 mM], and βME [3 mM]) at 4°C. The columns were washed with 10 column volumes (cv) of binding buffer, followed by 15 cvs of washing buffer (Tris [100 mM], pH 7.8, NaCl [250 mM], glycerol [10%], βME [3 mM], and imidazole [50 mM]). Recombinant luciferases were eluted by using elution buffer (Tris [100 mM], pH 7.8, NaCl [500 mM], glycerol [10%], βME [3 mM], and imidazole [300 mM]). Finally, the proteins were loaded onto a HiLoad Superdex S200 size-exclusion filtration column (GE Healthcare), attached to an AKTΔpurifier core system (GE Healthcare), and pre-equilibrated with filtration buffer (Hepes [20 mM], pH 7.8, NaCl [500 mM], glycerol [10%], and TCEP [1 mM]). The final protein peaks were collected and concentrated to ≈5–10 mg/ml (measured by absorbance) and their purity was analyzed by using SDS–PAGE.

## Crystallization

A high-throughput approach was used to explore initial crystallization conditions for RE$_{Ph}$ (7.2 mg/ml) and GB$_{Av}$ (5.6 mg/ml) luciferases that were previously purified. Commercially available conditions from screens PACT suite, JCSG (Qiagen), INDEX, and Crystal Screen 1 and 2 (Hampton Research), were tested by using a vapor diffusion method on sitting drops. A total of 75 µl of each condition was dispensed into a 96-well SD2 Molecular Dimensions (MD2) plate by a Bravo robot (Agilent Technologies). A volume of 250 nl of protein was mixed (1:1) with each condition by using a Honeybee X8 robot (Isogen Life Science). The plates were sealed, incubated at 18°C, and checked regularly under the microscope. Along with the free enzymes, several complexes that include the enzyme and ATP (in presence of MgCl$_2$), enzyme and luciferin, and enzyme and ATP and luciferin (supplemented with MgCl$_2$) were tested at different ligand concentrations (0.1–5.0 mM).

The hits obtained for both enzymes were systematically explored to improve the initial crystals by making 2D variations of the pH and all the components (precipitant and additives) in 96-well plates, and manually in MRC Maxi 48-Well Crystallization Plate (Swissci). The improved conditions were scaled to 1.5 µl drops (protein/precipitant = 2:1) and bigger single crystals were obtained for both GB$_{Av}$ and RE$_{Ph}$ (free enzymes). The final crystallization conditions were sodium citrate (1.6 M), pH 6.5, supplemented with MgCl$_2$ (200 mM) for GB$_{Av}$, and polyethylene glycol 3350 (19–21%), bis–tris propane (100 mM), pH 6.0, supplemented with ammonium sulfate (200 mM) for RE$_{Ph}$.

## X-ray data collection and structure determination

The crystals of both proteins were fast-frozen in liquid nitrogen. Only the crystals of RE$_{Ph}$ were soaked into a cryoprotectant solution (crystallization solution supplemented with 15–20% glycerol) as the final crystallization condition for GB$_{Av}$ crystals (sodium citrate [1.6 M], pH 6.5) worked as a cryoprotectant solution. The diffraction data were collected by using synchrotron radiation at the Swiss Light Source (beamline X06DA) with a Pilatus 2M detector (Dectris) and at the MX2 beamline from the Australian Synchrotron with a Quantum 315r Detector (ADSC). Routinely, 360° were collected for each crystal with an oscillation range between 0.1 and 0.5° and exposure time of 0.1–1 s per image, depending on the quality of crystals. Several datasets were collected for GB$_{Av}$ (free enzyme) up to 1.9 Å (Fig 1B), others for RE$_{Ph}$ (free enzyme) up to 3.1 Å at X06DA and up to 3.6 Å at MX2 beamline (Australian Synchrotron). Other datasets were collected for crystals of GB$_{Av}$ soaked with ATP (up to 2.2 Å). The crystals of RE$_{Ph}$ are in the triclinic space group *P*1 (parameters: *a* = 105.70 Å; *b* = 121.17 Å; *c* = 129.44 Å; *α* = 61.86°; *β* = 68.35°; *γ* = 74.17°) and trigonal *P*3$_1$21 (parameters: *a* = *b* = 119.100 Å; *c* = 351.402 Å; *α* = *β* = 90.00°; *γ* = 120.00°), whereas crystals of GB$_{Av}$ are in the orthorhombic space

group $P2_12_12_1$ (parameters: $a$ = 94.28 Å; $b$ = 110.53 Å; $c$ = 122.07 Å; $\alpha = \beta = \gamma = 90°$).

For the $P1$ crystal form of $RE_{Ph}$, two isomorphous datasets were merged to obtain a good dataset at 3.05 Å (Table S1), which was used to solve the structure. The datasets were indexed and integrated by using XDS (Kabsch, 2010) and iMosflm (Battye et al, 2011) and scaled with Scala (Evans, 2006) from the CCP4 program suite (Winn et al, 2011). The protein structures were phased by molecular replacement using the program Molrep (Vagin & Teplyakov, 1997) within the molecular replacement protocol of Auto-Rickshaw (Panjikar et al, 2005, 2009). The N-terminal domain of the Japanese firefly structure (PDB code: 2D1R) was used as a template. Refinement of the structures was carried out with the PHENIX program (Afonine et al, 2012). As a result of differences in the conformation of the C-terminal domains, this part of the structure was manually built in both molecules of $GB_{Av}$ found in the asymmetric unit, and later in the only complete monomer of $RE_{Ph}$ using the program Coot (Emsley et al, 2010). An iterative process of manual building and refinement with the PHENIX program (Afonine et al, 2012) was applied to both structures. Data processing and refinement statistics are summarized in Table S1. The final models presented good stereochemistry parameters and $R$ and $R_{free}$ values. Also, both $RE_{Ph}$ crystal forms showed good-quality electron density maps for their low resolution (Fig S1A and B). The figures of the structures were generated with PyMol (DeLano, 2002). Structure factors and atomic coordinates are deposited in the Protein Data Bank (PDB accession codes: 6AAA, 6ABH, and 6AC3).

## Differential scanning calorimetry (DSC)

Calorimetric analyses for the WT proteins and mutants were performed by using a Nano-DSC (TA Instruments). The samples and buffers were degassed under vacuum for 15 min with stirring at 10°C. The concentrations of all the proteins were adjusted to 0.5 mg/ml. A total volume of 300 $\mu$l of each degassed sample in Hepes (20 mM; pH 7.8) and $MgCl_2$ (10 mM) was loaded into the sample cell. For measurements with ligands, ATP and firefly luciferin (Gold Biotechnology) were added to both, the sample and reference cell in the same concentration (5 $\mu$M). The samples were heated at a scan rate of 1°C/min from 10 to 80°C at 3 atm. Before loading the samples, background scans were obtained by loading degassed buffer (with or without substrates) in both the reference and samples cells and heated at the same rate. The enthalpies of the transitions ($\Delta H_{cal}$) were estimated by calculating the area under the thermal transition after subtracting the blank and fitting a baseline by using Nano Analyzer software provided by the manufacturer.

## Computational analysis

The closed state of $GB_{Av}$ was modeled with classical MD and quantum mechanics (QM)/MM calculations. To integrate the oxyluciferin in its keto form and protonated adenosine 5′-monophosphate nucleotide (AMPH) molecules in the active site of $GB_{Av}$, manual docking was performed based on the position of these two molecules in the structures of the North American firefly, *P. pyralis* (PDB code: 4G37), and the Japanese firefly, *L. cruciata* (PDB code: 2D1R). All

calculations were performed on one of the molecules (chain B) in the $GB_{Av}$ structure.

The residues were protonated by using Leap from Amber14 suite of the program (Case et al, 2017). The contentious cases, especially for histidines, were resolved by computing their p$K_a$ with the H++ program (DeLano, 2002) to have a neutral charge for the system. Depending on the model considered, we selected to double-protonate (i.e., one hydrogen on each nitrogen of the side chain, which resulted in a positively charged residue) the following histidines for luciferases in complex with oxyluciferin and AMPH:

(1) $GB_{Av}$-open with, $GB_{Av}$-closed, and $GB_{Av}$-closed-I347L: the doubly protonated histidines are 5, 23, 42, 72, 305, 404, and 426.
(2) $GB_{Av}$-closed-R337L: the doubly protonated histidines are 5, 23, 42, 72, 305, 404, 426, and 456.
(3) $GB_{Av}$-closed-insert-R356: the doubly protonated histidines are 5, 23, 42, 72, 405, and 427.

AMPH was included with a single negative charge, and oxyluciferin was modeled in its phenolate-keto form, with a single negative charge.

Classical dynamics simulations were performed with Amber14 to obtain several snapshots for further QM/MM optimization. The model was solvated with TIP3P water molecules within a cubic box by ensuring a solvent shell of at least 15 Å around the solute. The resulting system contained ≈28,000 water molecules and 90,000 atoms in total. The AMBER99ff was used to model the residues of the protein. The AMPH and the emitter (oxyluciferin) were described by using parameters developed by the Navizet group (Navizet et al, 2010, 2011; Chen et al, 2011; Anandakrishnan et al, 2012). The parameters were not fully optimized for the excited state, so the oxyluciferin structure was first obtained by QM/MM optimization of the first singlet excited state and frozen in its excited state conformation during the whole dynamics simulation. The system was heated from 100 to 300 K in 20 ps. Then, under isothermal-isobaric ensemble (NPT) conditions with $T$ = 300 K and $P$ = 1 atm, a 10-ns dynamic with periodic boundary conditions was realized with a 2-fs time step. During these simulations, the pressure and temperature were maintained by using the Langevin algorithm with a coupling constant of 5 ps. SHAKE constraints were applied to all bonds that involved hydrogen atoms (Ryckaert et al, 1977). Random snapshots were extracted along the MD and used to compute the QM/MM emission. These snapshots correspond to low-energy points of the classical MD.

The folding of the C-terminal domain was performed by using umbrella sampling MDs (Kästner, 2011). In detail, the umbrella sampling was realized between the two $\alpha$-carbons of residues 311 and 524. The distance is 21 Å at the beginning and ends at 7 Å, with a step of 1 Å. For each step, an equilibration of 200 ps followed by a 750-ps production was realized. The lowest energy conformation was also collected for each step. This yields a set of structures along the path that decreases the distance between the two $\alpha$-carbons of residues 311 and 524. The bias introduced by the umbrella potential was removed by using the weighted histogram analysis method (Kumar et al, 1995; Roux, 1995), to generate a free-energy profile along the approach path.

The QM/MM calculations were performed by using a QM/MM coupling scheme between Gaussian (Frisch et al, 2016) and Tinker

(Tinker, 2005) (Gaussian_09d/Tinker). The electrostatic potential fitted method (Ferré & Ángyán, 2002) was used to compute the interaction between the Mulliken charges of the QM subsystem and the external electrostatic potential of the MM subsystem within 9 Å from the QM part. The microiterations technique (Melaccio et al, 2011) was used to converge the MM subsystem geometry for every QM minimization step. The emitter was selected as the QM subsystem, whereas the rest of the system was assigned to the MM subsystem. The QM/MM optimization of the first singlet excited state ($S_1$) was performed first, followed by calculation of the vertical difference of energies between $S_1$ and the ground state ($S_0$), which corresponds to the fluorescence emission. In fireflies, the fluorescence transition (induced by photoexcitation) is the same as the bioluminescence transition (obtained as a result of a bioluminescence reaction) (Navizet et al, 2013); therefore, the calculated emission value can be equated to the experimental emission energy.

The levels of theory (the treatment of the electron correlation and the basis set) used in the QM of the QM/MM calculations were chosen as follows: the time-dependent density functional theory (TD-DFT) calculations were carried out by using the B3LYP functional with the 6-311G(2d,p) basis set. The selected basis set was 6-311G(2d,p); we did not use basis set with diffuse functions because they can interact with the MM system with electrostatic embedding (electrostatic potential fitted method). As detailed in reference (Berraud-Pache & Navizet, 2016), these conditions are optimal for this type of calculation.

## Supplementary Information

## Acknowledgements

This work was supported by New York University (NYU) Abu Dhabi, the Research Enhancement Fund from NYU Abu Dhabi, and Abu Dhabi Education Council. This work was also financially supported by the Human Frontier Science Program (project RGY0081/2011, "Excited-State Structure of the Emitter and Color-Tuning Mechanism of the Firefly Bioluminescence"). The research was partially carried out using the Core Technology Platform resources at NYU Abu Dhabi. We thank the Swiss Light Source (beamline X06SA) and the Australian Synchrotron (beamline MX2) for providing the beamtime for X-ray data collection. I Navizet and R Berraud-Pache acknowledge support from the Agence Nationale de la Recherche (ANR) Biolum project (ANR-16-CE29-0013).

### Author Contributions

C Carrasco-López: data curation, formal analysis, investigation, methodology, project administration, writing—original draft, review, and editing.
JC Ferreira: data curation, formal analysis, and methodology.
NM Lui: data curation, formal analysis, and investigation.
S Schramm: data curation and formal analysis.
R Berraud-Pache: software and formal analysis.
I Navizet: software, formal analysis, writing—original draft, review, and editing.
S Panjikar: data curation and formal analysis.
P Naumov: conceptualization, funding acquisition, and writing—review and editing.
W Rabeh: supervision, funding acquisition, investigation, methodology, project administration, writing—original draft, review, and editing.

### Conflict of Interest Statement

The authors declare that they have no conflict of interest.

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
