## [Reviewer comments · Life Science Alliance]

Beetle luciferases with naturally red- and blue-shifted emission

César Carrasco-López, Juliana Ferreira, Nathan Lui, Stefan Schramm, Romain Berraud-Pache, Isabelle Navizet, Santosh Panjekar, Panče Naumov, and Wael Rabeih
DOI: 10.26508/lsa.201800072

Review timeline:

Submission Date:	23 April 2018
Editorial Decision:	31 May 2018
Revision Received:	26 July 2018
Editorial Decision:	01 August 2018
Accepted:	06. August 2018

Report:

(Note: Letters and reports are not edited. The original formatting of letters and referee reports may not be reflected in this compilation.)

May 31, 2018

Re: Life Science Alliance manuscript #LSA-2018-00072-T

Author information redacted

Dear Dr. Rabeh,

Thank you for submitting your manuscript entitled "Beetle luciferases with naturally red- and blue-shifted emission" to Life Science Alliance. The manuscript was assessed by three expert reviewers, whose comments are appended to this letter.

As you can see below, the manuscript received positive reviews overall. While all three referees do raise some concerns, there is also strong support and interest in your work. If you feel that you can address the concerns that have been raised, we would be happy to consider a revised version of your manuscript for publication in Life Science Alliance.

In considering a revised manuscript, we suggest focusing on the following items:

1. We feel that the comments from Reviewers 1 and 2 should be fully addressed, and providing electron density data and PDB codes is mandatory.
2. For Reviewer 3, please focus on comments 1, 2 and 4.
3. If it is possible to report specific activities, as indicated in comment 3 from Reviewer 3, please add these to the manuscript.
4. If crystal structures are now available for complexes with ligands (discussed by Reviewers 1 and 2), please consider adding these to the manuscript.

-- High-resolution figure, supplementary figure and video files uploaded as individual files: See our detailed guidelines for preparing your production-ready images, <http://life-science->

alliance.org/authorguide

B. MANUSCRIPT ORGANIZATION AND FORMATTING:

Full guidelines are available on our Instructions for Authors page, <http://life-science-alliance.org/authorguide>

Thank you for this interesting contribution to Life Science Alliance. We are looking forward to receiving your revised manuscript.

Sincerely,

Reviewer #1 (Comments to the Authors (Required)):

The manuscript describes the structures of luciferase enzymes from two species that have not previously been structurally characterized. The structures are used to inform mutagenesis studies to examine the role of several loops in influencing the wavelength of emitted light.

The structures include a red-shifted luciferase from *Phrixotrix hirtus* and a blue-shifted enzyme from *A. vivianii*. The structure of red-shifted RE(Ph) is at low resolution (3.0Å and 3.6Å), while the structure of the blue-shifted BG(Av) enzyme is at higher resolution (1.9Å). Both structures show high degrees of mobility of the C-terminal subdomain, a feature that has been observed previously in other luciferase structures, as well as other members of this broader enzyme family. Unfortunately, none of the structures contain bound ligands, a feature that also may relate to the high C-terminal subdomain disorder. The structural analysis is good; however the rather low resolution of the RE enzyme should be more fully discussed along with the potential limitations this provides for conclusions and analysis.

The use of organizational headings (Results, Discussion, etc..) would make it easier to read this manuscript.

Major concerns.

There is a great deal of discussion of the oligomeric state of the enzymes, including the octameric state of the RE enzyme observed in the P1 crystal. From the image, it appears this is simply a crystallographic packing of two tetramers. The authors should use the PISA informatics (through server or COOT) to suggest which of the subunit interactions are deemed to be biologically significant.

The disorder of the C-terminal subdomains has been observed before and should be noted more clearly that this is not unique to the current structures. Further, this raises the question of whether the "open" orientations observed are in fact real "states" or are simply positions where the crystal lattice stabilizes the position of the flexible sub-domain.

The paper ignores prior work that demonstrates the use of the second conformation for the oxidative reaction. This has been shown convincingly by both biochemical and structural work with the *P. pyralis* enzyme. On page 3, for example, the authors simply refer to "reversible opening and closing" without explaining that there are in fact two catalytic states. Further, the paper ignores some recent studies of red-shifting mutants of *P. pyralis*.

Pg 2 notes that "a single mutation, R11A, was sufficient to disrupt the octamer..." This is not shown by the data in figure S1c that shows that multiple mutations are required to shift the SEC peak to a monomer.

On middle paragraph of page 3, it is not clear what drove the authors to examine the 351-364 loop. Given the low resolution, how much confidence is there in the alternate positions observed. Further, how does this loop behave in alternate structures?

A statement (pg 4) like "We were able to discern with sufficient accuracy the structure of ..." suggests that the electron density here is likely weak. No density is shown in the manuscript and inclusion would allow the reader to judge for him or herself the quality of the data.

Minor concerns.

The authors should consider whether it is correct to use an adjective "N-terminal" to describe the domains. Either they are the N- and C-terminal domains or the N- and C-terminus. As the latter can refer simply to the two ends of the protein, I prefer including domains. The authors also use N-domain and C-domain on pg 2; that is acceptable also as long as they are consistent.

Table S2. Is the number of observations for the RE P1 merged structure (7.87 million) correct? For a low resolution structure that seems unlikely.

A sequence comparison, perhaps in the SI, for the structurally characterized proteins that inform the study should be considered. Additionally, this would be a good place to list the PDB codes for the structures being discussed from this paper and prior studies. The statement on page 3 that "RE is also the only beetle luciferase with an additional residue..." is not really conveyed by Figure 2e, as cited. An alignment would allow the reader to understand the context of the Arg insertion and would strengthen this statement. Further, a sequence alignment would give them an opportunity to label the catalytic lysine of the C-terminal domain, which would facilitate understanding of the discussion of residues K524, K526, and K529.

Supplemental Figure S1, the elution times of standards should be shown. Chromatographic traces for each do not need to be included.

No PDB accession codes are provided for the deposited structures.

Reviewer #2 (Comments to the Authors (Required)):

The article presents the first crystal structures of naturally red-emitting (REPh) and most blue-shifted green-emitting (GBAv) luciferases and the significance for two conserved loops in bioluminescent color determination. The bioluminescent color change is an enigma in bioluminescent research. The finding of the interaction of R337 with loop(351-364) in GBAv for green emission is very important. I feel regret that the crystal structure complexed with any ligands are not determined. However authors investigate well the effect of the active site microenvironment with structural, mutational, computational, thermal analysis in bioluminescent color determination. The article is well written and explains the findings in an understandable and appropriate way, and should be interest to the community.

Specific comments:

Sequence alignment of two loops with REPh, GBAv, GLc and GPp will assist in understanding your results.

Page 2, line 30 : "Supplementary Fig. 1d" is not in Supplementary Figures.

Page 2, line 42 : "RSMD" may be RMSD.

Page 3, line 10 : The authors should define "an angle between two domains".

S12 Sup Table1, REPh, No of reflections : The value 33877 is very low. The author will make a mistake in writing of the values of no. of reflections and unique reflections.

S12 Sup Table1, Refinement : Resolution and Rwork/Rfree need units of Å and %, respectively.

S18 Sup Fig1(a) : I can't understand the dashed lines and arrows.

Reviewer #3 (Comments to the Authors (Required)):

In this paper, for the first time, The crystal structures of two beetle luciferases with red- and blue-shifted light relative to the green-yellow light have been reported. They have shown that the structure of a blue-shifted green-emitting luciferase from the firefly *Amydetes vivianii* is monomeric with a structural fold similar to the previously reported firefly luciferases. The crystal structure of only known naturally red-emitting luciferase from the glow-worm *Phrixotrix hirtus* as tetramers and octamers.

1. The only main issue which should be taken to consideration is the real oligomer/monomer state. I am very keen to know if the protein has been formed in dimer or tetramer form upon purification or its real structural base.
2. A SDS-PAGE and PAGE image of purified protein could be crucial.
3. Specific activity of the enzyme compared to other reported firefly luciferases like *P.pyralis* or *L.turkestanicus* should be reported if is available.
4. There are more reports on the role of flexible loop of 352-364 in its critical role in water accessibility to the luciferin binding site. Is is in line with your investigation?
There are some minor language problem which should be corrected .

Review from referee 1.

The manuscript describes the structures of luciferase enzymes from two species that have not previously been structurally characterized. The structures are used to inform mutagenesis studies to

examine the role of several loops in influencing the wavelength of emitted light.

The structures include a red-shifted luciferase from *Phrixotrix hirtus* and a blue-shifted enzyme from *A. vivianii*. The structure of red-shifted RE_(Ph) is at low resolution (3.0Å and 3.6Å), while the structure of the blue-shifted BG_(Av) enzyme is at higher resolution (1.9Å). Both structures show high degrees of mobility of the C-terminal subdomain, a feature that has been observed previously in other luciferase structures, as well as other members of this broader enzyme family.

Unfortunately, none of the structures contain bound ligands, a feature that also may relate to the high C-terminal subdomain disorder. The structural analysis is good; however the rather low resolution of the RE enzyme should be more fully discussed along with the potential limitations this provides for conclusions and analysis.

We included in different parts of the manuscript the fact that RE_{Ph} was acquired at low resolution. Although the resolution is low, a high-quality electron density maps were acquired for the *P1* and *P3*:21 crystal forms of the RE_{Ph}. Supplementary Fig. 1 shows different electron density maps and

we added the following text to emphasize the low resolution of RE_{Ph} structures in the manuscript.

Page 2: “The crystal structure of WT RE_{Ph} was determined at low resolution by molecular replacement from two different crystal forms in the space groups *P1* and *P3₁21* at resolution of 3.05 Å and 3.60 Å, respectively (Supplementary Table 1).”

Page 2: “With the low resolution of the RE_{Ph} structure, site directed mutagenesis clearly confirms the interface interactions, where the single mutation, R11A, was sufficient to disrupt the octamer of RE_{Ph} and resulted exclusively in monomers in solution (Supplementary Fig. 2c).”

Page 3: “The loop^{348–361} in RE_{Ph}, albeit at low resolution, was modeled in the electron-density map (Supplementary Fig. 1c).”

Page 4: “Despite the low resolution of the RE_{Ph} structure, the high-quality electron density map allowed to model this loop region in the *P3₁21* crystal form (Supplementary Fig. 1d), where.....”

The use of organizational headings (Results, Discussion, etc...) would make it easier to read this manuscript.

We have included the headings as suggested.

Major concerns.

- 1) There is a great deal of discussion of the oligomeric state of the enzymes, including the octameric state of the RE enzyme observed in the P1 crystal. From the image, it appears this is simply a crystallographic packing of two tetramers. The authors should use the PISA informatics (through server or COOT) to suggest which of the subunit interactions are deemed to be biologically significant.

The oligomeric state of the wild-type (WT) RE_{Ph} has been addressed extensively and confirmed through site directed mutagenesis and gel filtration analysis. These give a strong evidence that the WT RE_{Ph} is an oligomer in solution. As pointed here by the reviewer, the RE_{Ph} in the P1-crystal arrangement is a dimer of tetramers, with the tetramers being the relevant biological conformation or at least the most stable in the *in vitro* conditions used here. The octamer with dimer of tetramers arrangement could also be concentration dependent, where the concentrations used in the crystallization trial might be similar to the biological concentration in the railroad worm.

The PISA analysis did not yield an assembly that could be stable in solution. Nevertheless, one of the main factors taken into consideration by PISA calculations are water accessible surfaces, which are determined by the interactions of water molecules present in the structure. The absence of waters in the RE_{Ph} structure due to the low resolution is introducing a bias in the PISA calculations.

- 2) The disorder of the C-terminal subdomains has been observed before and should be noted more clearly that this is not unique to the current structures. Further, this raises the question of whether the "open" orientations observed are in fact real "states" or are simply positions where the crystal lattice stabilizes the position of the flexible sub-domain.

We added three citations for manuscripts showing the high flexibility in the C-terminal domains of firefly luciferases from *Photinus pyralis*, where the C-terminal domain was not modeled due to lack of a defined electron density map. This indicates the high flexibility of the C-terminal domain in these structures. The following sentence was added to the manuscript:

Page 2: “The inability to model the C-terminal domain of firefly luciferases in certain crystal conditions, as result of its high flexibility, has been previously shown (Auld et al, 2010; Thorne et al, 2012; Kheirabadi et al, 2013).”

In addition, the two conformations of the C-terminal domain we observed in the GB_{Av} have not been determined previously according to the structural alignment with all the available structures from beetle luciferases. The two conformations of the C-terminal domain of GB_{Av} are structurally positioned between the previously determined open (PDB: 1LCI and 5DV9) and closed (PDB: 2D1R) states without forcing the structures with the introduction of Cys-Cys disulfide bonds. The two conformations of the C-terminal domain may or may not be biologically relevant, but it is another evidence of the C-terminal domain flexibility, where these conformations are the most stable and are able to crystallize.

- 3) The paper ignores prior work that demonstrates the use of the second conformation for the oxidative reaction. This has been shown convincingly by both biochemical and structural work with the *P. pyralis* enzyme. On page 3, for example, the authors simply refer to "reversible opening and closing" without explaining that there are in fact two catalytic states. Further, the paper ignores some recent studies of red-shifting mutants of *P. pyralis*.

Thanks for making us aware of the two catalytic conformations. We indicated the two catalytic conformations in the manuscript:

Page 3: "Together with the structural data, these results confirm the pronounced mobility of the C-terminal domain of beetle luciferases, which is capable of reversible opening and closing of the active site through two catalytic conformations during the bioluminescence reaction. The two catalytic conformations are stimulated by rotation on the C-terminal domain of firefly luciferases (Sundlov, et al, 2012)."

- 4) Pg 2 notes that "a single mutation, R11A, was sufficient to disrupt the octamer..." This is not shown by the data in figure S1c that shows that multiple mutations are required to shift the SEC peak to a monomer.

The size-exclusion chromatography (SEC) data for R11A of RE_{Ph} has been added as Supplementary Fig. 2c, the new panel is shown below. The estimated size of the single mutant R11A of RE_{Ph} is similar to the monomeric GB_{Av} luciferase. The single mutant R11A and the triple mutant R11A, Y153A, and F162A of RE_{Ph} yielded similar retention time as the monomeric GB_{Av} luciferase (Supplementary Fig. 2c). The data shown here support our statement that the introduction of the single mutant R11A was sufficient to disrupt the oligomeric packing of RE_{Ph} and produced monomers.

- 5) On middle paragraph of page 3, it is not clear what drove the authors to examine the 351-364 loop.

Given the low resolution, how much confidence is there in the alternate positions observed. Further, how does this loop behave in alternate structures?

Several reasons drove us to this region, starting by the insertion of Arg 353, previously shown to be important in the Red shift the emission in green luciferases and also its proximity to the Arg337, which is one of the most remarkable substitution observed in RE_{Ph}. As discussed in the manuscript, the loose packing of loop^{348–361} in the N-terminal domain of RE_{Ph} in comparison to the tight packing of the same loop^{351–364} in GB_{Av} pointed us to explore the role of this loop in the color tuning mechanism (Figure 2). Comparison of loop^{351–364} in GB_{Av} with the same loop in *Photinus pyralis*, *Luciola cruciate*, and *Lampyris turkestanicus* reveal minimum conformational changes between these structures (Supplementary Fig. 6b – 6d). However, a large conformational movement of loop^{348–361} in RE_{Ph} by ~12Å is observed in comparison to GB_{Av} that is due to the substitution of Arg 337 and Glu 354 in GB_{Av} with Leu 334 and Asn 351 in RE_{Ph}, respectively (Fig. 2e and 2f). Biochemical analysis and site directed mutagenesis support the importance of this loop in the color tuning mechanism.

The following sentence have been included to clarify the interest in investigating the importance of this loop in the color tuning mechanism:

Page 3: “The presence of the only known insertion in beetle luciferases (R353 in RE_{Ph}), its proximity to both the active site and several key substitutions around the benzothiazole moiety that were previously found to have an impact in the color tuning (Viviani et al, 2007), drove us to further investigate its relevance in the color-tuning mechanism.”

Below is Supplementary Fig. 6 that has been generated to show the conformational differences of loop^{351–364} in RE_{Ph} in comparison to the same loop in GB_{Av} and other green emitting luciferases.

6) A statement (pg 4) like "We were able to discern with sufficient accuracy the structure of ..."

suggests that the electron density here is likely weak. No density is shown in the manuscript and inclusion would allow the reader to judge for him or herself the quality of the data.

Supplementary fig. 1 has been added to show the general quality of the data and electron density maps, particularly for the loops mentioned in the manuscript.

Minor concerns.

- 1) The authors should consider whether it is correct to use an adjective "N-terminal" to describe the domains. Either they are the N- and C-terminal domains or the N- and C-terminus. As the latter can refer simply to the two ends of the protein, I prefer including domains. The authors also use N-domain and C-domain on pg 2; that is acceptable also as long as they are consistent.

The text has been corrected to reflect the N-terminal domain or C-terminal domain throughout the manuscript.

- 2) Table S2. Is the number of observations for the RE P1 merged structure (7.87 million) correct? For a low resolution structure that seems unlikely.

This is a typo, and the correct number of total reflections is 517,157.

- 3) A sequence comparison, perhaps in the SI, for the structurally characterized proteins that inform the study should be considered. Additionally, this would be a good place to list the PDB codes for the structures being discussed from this paper and prior studies. The statement on page 3 that "RE is also the only beetle luciferase with an additional residue..." is not really conveyed by Figure 2e, as cited. An alignment would allow the reader to understand the context of the Arg insertion and would strengthen this statement. Further, a sequence alignment would give them an opportunity to label the catalytic lysine of the C-terminal domain, which would facilitate understanding of the discussion of residues K524, K526, and K529.

The sequence alignment figure has been added as Supplementary Figure 5.

- 4) Supplemental Figure S1, the elution times of standards should be shown. Chromatographic traces for each do not need to be included.

The included SEC chromatograms are only for mutants that changed the oligomerization state of the RE_{Ph} in addition to the WT GB_{Av} that has been used as a reference for the monomeric structures. The included traces are important to show the different oligomerization states of RE_{Ph} mutant enzymes that disrupted the packing of the oligomerization state of the WT RE_{Ph}.

- 5) No PDB accession codes are provided for the deposited structures.

The structures are submitted to the PDB and the following accession codes have been assigned: 6AAA, 6ABH and 6AC3 (Supplementary Table 1).

Review from referee 2.

The article presents the first crystal structures of naturally red-emitting (RE_{Ph}) and most blue-shifted green-emitting (GB_{Av}) luciferases and the significance for two conserved loops in bioluminescent color determination. The bioluminescent color change is an enigma in bioluminescent research. The finding of the interaction of R337 with loop(351-364) in GB_{Av} for green emission is very important. I feel regret that the crystal structure complexed with any ligands are not determined. However authors investigate well the effect of the active site microenvironment with structural, mutational, computational, thermal analysis in bioluminescent color determination. The article is well written and explains the findings in an understandable and appropriate way, and should be interest to the community.

Specific comments:

- 1) Sequence alignment of two loops with RE_{Ph}, GB_{AV}, G_{LC} and G_{PP} will assist in understanding your results.
The sequence alignment figure has been added as Supplementary Figure 5.
- 2) Page 2, line 30 : "Supplementary Fig. 1d" is not in Supplementary Figures.
Thanks for pointing this out. It should be Supplementary Fig. 2c. It has been changed in the manuscript.
- 3) Page 2, line 42 : "RSMD" may be RMSD.
Thanks for pointing this out. It has been corrected to RMSD.
- 4) Page 3, line 10: The authors should define "an angle between two domains".
The angle here refers to the aperture of the active site, considering axes crossing the center mass of each domain. The angle has been defined in the sentence as follow:
Page 3: "The aperture of the active site with an angle of ~ 125° between axes crossing the center mass of each domain (P3₁21 crystal form) shows RE_{Ph} is the most open conformation among the known luciferase structures, which corresponds to the high mobility of its C-terminal domain."
- 5) S12 Sup Table1, RE_{Ph}, No of reflections : The value 33877 is very low. The author will make a mistake in writing of the values of no. of reflections and unique reflections.
Thank for pointing this out. The numbers of reflections and unique reflections were inverted. The correct values are 186,010 for the number of reflections and 33,877 for the unique reflections. Taking into consideration the low resolution (3.6Å), these numbers are accurate.
- 6) S12 Sup Table1, Refinement: Resolution and R_{work}/R_{free} need units of Å and %, respectively.
The units are included in the Sup. Table 1.
- 7) S18 Sup Fig1(a) : I can't understand the dashed lines and arrows.
The following sentence has been added to the figure legend to clarify the use of lines and arrows.
Supplementary Fig. 2a: "The dotted lines separate the N-terminal domains from the C-terminal domains in the complex and the arrows indicate the direction of the movement of the C-terminal domains during catalysis."

Review from referee 3.

In this paper, for the first time, the crystal structures of two beetle luciferases with red- and blue-shifted light relative to the green-yellow light have been reported. They have shown that the structure of a blue-shifted green-emitting luciferase from the firefly *Amydetes vivianii* is monomeric with a structural fold similar to the previously reported firefly luciferases. The crystal structure of only known naturally red-emitting luciferase from the glow-worm *Phrixotrix hirtus* as tetramers and octamers.

1. The only main issue which should be taken to consideration is the real oligomer/monomer state. I am very keen to know if the protein has been formed in dimer or tetramer form upon purification or its real structural base.
The WT RE_{Ph} elute as an oligomer from size exclusion chromatography (SEC) with molecular weight equivalent to the tetramer in solution. The introduction of mutation in the tetramer interface produced only dimers based on SEC analysis (Supplementary Fig. 2c). However, the single mutant, R11A, in the dimer interface was sufficient to disrupt the interactions in both interfaces to produce exclusively monomers (the SEC profile of R11A mutant has been added to panel 1c).

Based on the SEC analysis of the RE_{Ph} mutants, the interactions in the dimer interface are stronger than those in the tetramer interface, as a result of the polar nature of the interactions in the dimer interface in comparison to weaker hydrophobic interactions in the tetramer interface.

2. A SDS-PAGE and PAGE image of purified protein could be crucial.

The SDS-PAGE of all proteins shown in Supplementary Fig. 2c has been added as Supplementary Fig. 2d. All proteins purified in this manuscript have purity >90% based on Coomassie-staining.

3. Specific activity of the enzyme compared to other reported firefly luciferases like *P.pyralis* or *L. turkestanicus* should be reported if is available.

The specific activities of these enzymes have been previously reported and since the kinetic analysis on the luciferase reaction will not improve the understanding of the color emission mechanism, we did not perform kinetic measurements in this manuscript as the main focus was on the understanding of the fine color-tuning mechanism.

4. There are more reports on the role of flexible loop of 352-364 in its critical role in water accessibility to the luciferin binding site. Is it in line with your investigation?

This is a very important finding in our manuscript as we strongly believe that the microenvironment of the active site is directly related to the color emission of beetle luciferases. As a result, the amount of water in the active site can be one of the factors that affect the active site microenvironment in addition to the charge of the amino acids occupying the active site. Unfortunately, due to the low resolution of the RE_{Ph} structure no water molecules could be modeled to support/refute previous findings.

There are some minor language problem which should be corrected.

August 1, 2018

RE: Life Science Alliance Manuscript #LSA-2018-00072-TR

Prof. Wael Rabeh
New York University Abu Dhabi
Sciences
PO Box 129188
Sadyaat Island, Abu Dhabi 129188
United Arab Emirates

Dear Dr. Rabeh,

Thank you for submitting your revised manuscript entitled "Beetle luciferases with naturally red- and blue-shifted emission". As you will see, the reviewers are pleased with the performed revisions, and we would be happy to publish your paper in Life Science Alliance pending final revisions necessary to meet our formatting guidelines.

Please proofread your manuscript one more time to address the comment made by reviewer #1. Please also mention all figures and S figures and their individual panels in the manuscript text (currently missing eg: Fig4, SFig6, 8, 9 and 10 and STable 3, 5, and 6 as well as some individual panels). Furthermore, please provide all S Tables and S figures as individual figure files. The current suppl docx file can remain, please simply upload the figures in an appropriate format in addition.

A. FINAL FILES:

-- High-resolution figure, supplementary figure and video files uploaded as individual files: See our detailed guidelines for preparing your production-ready images, <http://life-science-alliance.org/authorguide>

B. MANUSCRIPT ORGANIZATION AND FORMATTING:

Full guidelines are available on our Instructions for Authors page, <http://life-science-alliance.org/authorguide>

Sincerely,

Reviewer #1 (Comments to the Authors (Required)):

The revised manuscript has addressed my prior concerns. There are still a handful of typographical errors that the authors should address.

Reviewer #2 (Comments to the Authors (Required)):

The authors sufficiently answered my queries. There are no additional issues to be addressed.

Reviewer #3 (Comments to the Authors (Required)):

The manuscript is sufficiently revised and can be accepted.

August 6, 2018

RE: Life Science Alliance Manuscript #LSA-2018-00072-TRR

Prof. Wael Rabeh
New York University Abu Dhabi
Sciences
PO Box 129188
Sadyaat Island, Abu Dhabi 129188
United Arab Emirates

Dear Dr. Rabeh,

Thank you for submitting your Research Article entitled "Beetle luciferases with naturally red- and blue-shifted emission". It is a pleasure to let you know that your manuscript is now accepted for publication in Life Science Alliance. Congratulations on this interesting work.

The final published version of your manuscript will be deposited by us to PubMed Central (PMC) as soon as we are allowed to do so, the application for PMC indexing has been filed. You may be eligible to also deposit your Life Science Alliance article in PMC or PMC Europe yourself, which will then allow others to find out about your work by Pubmed searches right away. Such author-initiated deposition is possible/mandated for work funded by eg NIH, HHMI, ERC, MRC, Cancer Research UK, Teletthon, EMBL.

Please also see:

<https://www.ncbi.nlm.nih.gov/pmc/about/authorms/>

<https://europepmc.org/Help#howsubsmanu>

*****IMPORTANT:** If you will be unreachable at any time, please provide us with the email address of an alternate author. Failure to respond to routine queries may lead to unavoidable delays in publication.*******

DISTRIBUTION OF MATERIALS:

Again, congratulations on a very nice paper. I hope you found the review process to be constructive and are pleased with how the manuscript was handled editorially. We look forward to future exciting submissions from your lab.

Sincerely,
